# Influences of Residual Stress, Surface Roughness and Peak-Load on Micro-Cracking: Sensitivity Analysis

Jairan Nafar Dastgerdi [1,2,*] , Fariborz Sheibanian [1], Heikki Remes [2] and Hossein Hosseini Toudeshky [1]

1 Department of Aerospace Engineering, Amirkabir University of Technology, 424 Hafez Avenue, Tehran 15875-4413, Iran; f.sheibanian@aut.ac.ir (F.S.); hosseini@aut.ac.ir (H.H.T.)
2 Department of Mechanical Engineering, School of Engineering, Aalto University, P.O. Box 14300, Aalto, FIN-00076 Espoo, Finland; heikki.remes@aalto.fi
* Correspondence: jairan.nafardastgerdi@aalto.fi; Tel.: +98-21-6454-5636

**Abstract:** This paper provides further understanding of the peak load effect on micro-crack formation and residual stress relaxation. Comprehensive numerical simulations using the finite element method are applied to simultaneously take into account the effect of the surface roughness and residual stresses on the crack formation in sandblasted S690 high-strength steel surface under peak load conditions. A ductile fracture criterion is introduced for the prediction of damage initiation and evolution. This study specifically investigates the influences of compressive peak load, effective parameters on fracture locus, surface roughness, and residual stress on damage mechanism and formed crack size. The results indicate that under peak load conditions, surface roughness has a far more important influence on micro-crack formation than residual stress. Moreover, it is shown that the effect of peak load range on damage formation and crack size is significantly higher than the influence of residual stress. It is found that the crack size develops exponentially with increasing peak load magnitudes.

**Keywords:** surface roughness; residual stresses; peak load; finite element method; micro-crack formation

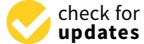



## 1. Introduction

The effect of surface treatments and different processing methods on the performance of high-strength steel under high peak stresses, either as single events or as a part of service loading, is the main concern of strength design and life estimation. The main parameters affecting and describing surface integrity are surface roughness, residual stress, and the material properties in the surface layer. These parameters can vary separately as a result of the manufacturing procedures and machining conditions [1–3]. Therefore, the influences of these affecting parameters on the performance and failure damage mechanism of high-strength steels in real engineering applications should be investigated.

It has long been identified that fatigue cracks generally nucleate from the free surface and the local microscopic stress and strain concentration at the surface defects are remarkable factors for crack nucleation and propagation [4–8]. Surface roughness is a significant index for characterizing surface micro-topography, and it has a critical effect on fatigue life [9,10]; thus, the influence of surface roughness on fatigue performance has been a comprehensive research area for several years, and numerous studies have investigated the effects and function mechanism of surface roughness caused by different surface processing methods on the fatigue behavior of diverse materials [11–15]. Despite significant advances in understanding the effect of surface roughness on the fatigue performance of various kinds of materials, the role of surface roughness in crack nucleation and the damage mechanism under peak load, e.g., as an initial step before fatigue loading, is poorly understood. Moreover, the influences of residual stress and material properties on the surface layer are still obscure, considering the simultaneous impact of surface roughness on crack formation and failure mechanism.

However, some researchers have experimentally studied the effect of surface integrity on the fatigue strength of high-strength steels [16–19]. They have provided a comprehensive overview of the influence of these parameters on fatigue strength, although there is still an insufficient understanding of the effects of surface roughness, hardness, and residual stress on failure mechanism, crack initiation, and propagation under loading for high-strength steel materials. Therefore, predicting the structure strength and mechanical performance under loading with new manufacturing processes and machining parameters is not possible except by performing new time-consuming and expensive tests. A better model, encompassing the detailed characteristics of the surfaces with all affecting parameters, is expected to capture the failure mechanism over a broad range of conditions. Thus, it is imperative to study the combined effect of surface integrity parameters on the damage mechanisms and micro-cracking of engineering structures and components under peak loads, which is the focus of this investigation.

Ductile fracture has been extensively investigated for the purpose of modeling and assessing the failure mechanisms of materials and structures, especially in the context of metals and alloys used in different engineering practices using various proposed ductile fracture criteria [20–23]. These criteria have been extended based on various assumptions, hypotheses, and experimental observations related to ductile fracture. The phenomenological ductile fracture criterion has been used to develop models for the prediction of fracture processes such as crack nucleation, propagation and failure mechanisms [24–26]. To implement the ductile fracture criterion into surface integrity analysis, a new finite element (FE) modeling approach was recently developed by the authors. The developed approach fully captures the complexity of the surface roughness using actual two-dimensional surface topography and the effect of residual stress by introducing global layer-wise modeling, with a constant temperature in each layer following the measured residual distribution [27].

In this study, the micro-mechanism motivated phenomenological damage model was applied to predict ductile fracture initiation in the context of stress triaxiality and equivalent plastic strain [28]. The ductile fracture criterion has been calibrated for the prediction of fracture locus with an inverse numerical-experimental approach in order to determine the material constants in the criterion [27]. Although the approach is successfully calibrated with experiments, it has not yet been applied for the systematic analysis of surface integrity effects. Since surface integrity includes several influencing factors, in addition to experimental observation, numerical simulation is necessary to reveal the main affecting factors. For instance, compressive loads tend to relax compressive residual stresses in proportion to their magnitude. Relaxation due to a single peak load is usually evaluated as being of greater significance than gradual cyclic relaxation [29,30]. As residual stress relaxation pertains to the correlation of the local stresses and the local yield strength [31], surface roughness as an affecting parameter on the local stress concentration influences relaxation behavior. It is unclear under what conditions residual stress relaxation arises, or what the impact of residual stress distribution on crack formation and damage mechanism may be. It is well known that residual stress relaxation is a complex phenomenon [32]. Thus, the integration of residual stress in predictive modeling computation, without consideration of relaxation during operation, results in an imprecise prediction for the trustworthiness and reliability of the components and structures. It is worthwhile mentioning that this study is the second part of the authors' recent work [27], which provides a further understanding of the influence of compressive peak load on the micro-crack formation, crack size, and residual stress state of sandblasted high-strength steels in real engineering applications when peak load is applied to the surface before or during fatigue loading. The influence of peak load on residual stress distribution was studied, and further analysis was carried out to investigate the effects of peak load, affecting the parameters of fracture locus, and residual stress, affecting damage mechanism and crack size.

## 2. Materials and Methods

### 2.1. Characteristics of Sandblasted High-Strength Steel Surface

This study focuses on 15-mm-thickness sandblasted 690 high-strength steel plate. In this case, previous experimental investigations provide good descriptions of roughness measurement, residual stress, and material properties [27]. The surface roughness measurements were carried out according to SFS-EN ISO 4288 [33], and the size of the surface profiles was defined for the rolled plate. The surface contour is depicted in Figure 1. The arithmetical average value, $R_a$, and the average of the five largest peak-to-peak values, $R_z$, were identified for this surface contour. These values were 128 μm and 235 μm, respectively. The elastic properties of the material are described using a Young's modulus of E = 210 GPa, and a Poisson ratio of $\nu$ = 0.3. The chemical composition of the studied material is presented in Table 1. Due to the complexity of the surface topology, high local stresses are expected to arise at the surface. To evaluate the effect of local plasticity on the near surface stress fields, the von Mises yield criterion is used in the simulations, assuming associated plastic flow and isotropic hardening.

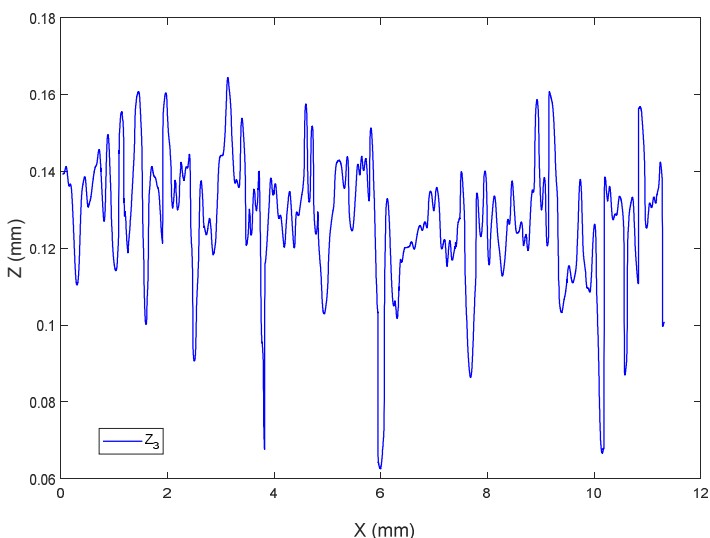

**Figure 1.** Profile of the surface roughness [27].

**Table 1.** Chemical composition (%). Reprinted with permission from ref. [18]. Copyright 2021 Springer Nature.

| Material | C | Si | Mn | P | S | Al | Nb | V | Ti | Cu | Cr | Ni | Mo | B | N |
|---|---|---|---|---|---|---|---|---|---|---|---|---|---|---|---|
| S690 | 0.16 | 0.21 | 1.39 | 0.011 | 0.001 | 0.047 | 0.015 | 0.018 | 0.007 | 0.01 | 0.25 | 0.06 | 0.502 | 0.001 | 0.002 |

In this study, the through-thickness residual stress distribution with the maximum stress value (−80 MPa) measured at the surface [27] is employed. This distribution is obtained from the strain-gage hole drilling measurements for sandblasted steel samples [32]. Furthermore, to investigate the compressive residual stress effect on micro-crack formation and crack size, the residual stress value at the surface is increased from −80 MPa to −320 MPa, as reported experimentally for the grinding manufacturing process of 690 high-strength steel plate [1].

### 2.2. Ductile Fracture Criterion

In this simulation, the micro-mechanism-inspired phenomenological damage model described in [28], which uses a fracture strain that is dependent on stress triaxiality, is employed for the evolution of damage. The damage model and calibration approach is briefly presented in this section in order to describe the physical basis for and the parameters that affect the numerical simulations carried out in this paper. The calibration approach

employed, which is a new and efficient method, was explained in details in Ref. [27] by the authors.

The criterion is constructed according to the damage accumulation induced by the nucleation, growth and shear coalescence of voids [28], which can be written as follows:

$$\left(\frac{2\tau_{\max}}{\overline{\sigma}}\right)^{C_1} \times \left(\frac{\langle 1 + 3\eta \rangle}{2}\right)^{C_2} \times \overline{\varepsilon_f} = C_3 \quad \langle x \rangle = \left\{ \begin{array}{ll} x, & \text{when } x \geq 0 \\ 0, & \text{when } x < 0 \end{array} \right. \tag{1}$$

The model is founded based on the microscopic analysis of ductile fracture, where void nucleation is explained as a function of the equivalent plastic strain, the void growth is demonstrated as a function of stress triaxiality, as $1 + 3\eta$, and the coalescence of voids is represented by the normalized maximal shear stress, defined as $\tau_{\max}/\overline{\sigma}$. The influence of nucleation, growth, and coalescence of voids is controlled by the two calibration exponents: $C_1$ and $C_2$. The material constant $C_3$ is equal to the equivalent plastic strain required to fracture in uniaxial tension. This model is able to explain these various phenomena, and the shape of the fracture locus can be easily controlled using the three material constants. Thus, it is preferable to use this criterion. The aforementioned equation gives the fracture strain for the full range of stress triaxiality.

Ideally, the input fracture criterion $\overline{\varepsilon_f}$ in Equation (1) should be specified based on tests covering the full range of stress triaxiality. For instance, in the study in which the fracture model was introduced [28], the material constants $C_1$, $C_2$ and $C_3$ were determined by fitting a curve to the experimental data. The same approach is applied in this study, but only uniaxial tension test data, together with single peak tensile load, are used. The input fracture criterion in the uniaxial tension range is calibrated based on the equivalent plastic strain computed at fracture initiation, failure strain $\overline{\varepsilon_f}$. The $\overline{\varepsilon_f}$ at uniaxial tension $\eta = 1/3$ gives the value for material constant $C_3$. This is calculated using an approach that first computes the true stress-strain curve of S690 high-strength steel under uniaxial loading. Then, the defined true stress-strain curve is implemented in the FE simulation using a UMAT subroutine to compute the equivalent von Mises plastic strain at fracture initiation. The algorithm of this approach is explained in detail in Reference [27]. In this study, the numerical simulation is carried out with Abaqus version 6.14 (Dassault Systèmes Simulia Corporation, Johnston, RI, USA).

To provide a full-range true stress-strain ($\sigma_t - \varepsilon_t$) curve for S690 high-strength steel under uniaxial loading, the instantaneous area method is applied to analyze the data provided using a digital imaging correlation technique. This technique has been employed to measure deformation fields of steel coupons during the whole range of deformation. Using this approach, the constitutive model for S690 high-strength steel materials is introduced, and the true strain is computed as follows [34]:

$$\sigma_t(\varepsilon) \left\{ \begin{array}{ll} \sigma_t = E\varepsilon_t & \text{for } \varepsilon_t \leq \varepsilon_y \\ \sigma_t = S_y \times \left[1 + 3 \times 10^{-3} \times \left(\frac{\varepsilon_t}{\varepsilon_y} - 1\right)\right] & \text{for } \varepsilon_y \leq \varepsilon_t \leq 6\varepsilon_y \\ \sigma_t = S_y \times \left\{1.015 + 0.1 \times \left[1 - 0.01 \times \left(0.6\left(\frac{\varepsilon_t}{\varepsilon_y} - 6\right) - 10\right)^2\right]\right\} & \text{for } 6\varepsilon_y \leq \varepsilon_t \leq 15\varepsilon_y. \\ \sigma_t = S_y \times \left\{1.094 + 0.1 \times \left(\frac{\varepsilon_t/\varepsilon_y - 15}{100}\right)^{0.45}\right\} & \text{for } 15\varepsilon_y \leq \varepsilon_t \leq 130\varepsilon_y. \\ \sigma_t = S_y \times \left\{1.2 - 0.09 \times \left[\left(\frac{\varepsilon_t}{\varepsilon_y} - 130\right)^{1.1}/340\right]\right\} & \text{for } 130\varepsilon_y \leq \varepsilon_t \leq 1.2. \end{array} \right. \tag{2}$$

where $E = 210$ MPa and $S_y = 770$ MPa is the yield strength of the material, and $\varepsilon_y$, $\varepsilon_t$ are the strain at yielding and fracture, respectively. In this study, the ductile fracture criterion is employed for damage initiation, and a fracture energy base criterion is applied for damage evolution.

Using the proposed approach, the value of the material constant $C_3$ is 1.2, which gives a $\overline{\varepsilon_f}$ at uniaxial tension of $\eta = 1/3$. Then, various values for $C_1$, $C_2$ were considered on the basis of the common range ($1 < C_1 < 8$ and $0 < C_2 < 1$) for these parameters [28]. Employing an inverse numerical–experimental approach proposed by the authors, $C_1$,

$C_2$ can be calibrated in such a way that FE simulation for pre-notched specimens with ductile damage can predict the micro-cracks observed from SEM images in the proximity of the notch under peak load conditions [27]. The material constant $C_1$ modulates the effect of the normalized maximal shear stress on the shear coalescence of voids during plastic deformation. As $C_1$ increases, the influence of the maximal shear stress on ductile fracture increases, and accordingly, the fracture strain at $\eta = 0$ decreases. The sensitivity analyses for investigating the effects of the material constants $C_1$ on the ductile fracture criterion and micro-crack formation reveal that damage-induced micro-crack formation is insensitive to the values of $C_1$ under compressive peak load conditions [27]. A common mean value, $C_1 = 4$, is observed to give a reliable estimation for surface integrity analysis.

For negative stress triaxialities, the fracture is governed by the shear mode. Power exponent $C_2$ is added to the void growth function in order to represent void coalescence. Therefore, if parameter $C_2$ increases, void growth in compression will be greatly suppressed, and the progress of void growth under shear stress will be slower, with void coalescence being distinctly compressed. Since the surface has negative residual stress and micro-defects and is subjected to the compressive overload, the damage occurs mainly on the compression side of the fracture locus, and thus, $C_2$ is the most influential parameter on micro-crack formation, as depicted in Figure 2. This figure shows the calibrated fracture locus for various values of $C_2$. Employing an inverse numerical–experimental approach, the correct value for the constant $C_2$ is selected based on the micro-crack length obtained from microscopic analysis of the surface. When the micro-crack length is known, the calibration of the constants $C_2$ is easy, since the length of the formed micro-cracks is sensitive to the constant $C_2$.

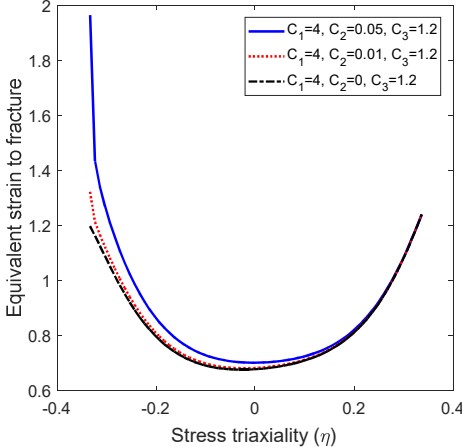

**Figure 2.** Fracture locus for high-strength steel 690.

This new special approach is employed in this study, and the fracture locus constructed using this calibration approach with the material constants $C_1 = 4$, $C_2 = 0.01$, $C_3 = 1.2$ is further used in the FE simulation, as shown in Figure 2. Moreover, a sensitivity analysis is carried out on the effect of $C_2$ as the most influential parameter with respect to micro-crack formation and crack length under compressive overload conditions using different fracture locus.

### 2.3. Numerical Simulation

The FE model was built using the two-dimensional real surface contour of the specimen measured by profilometry to capture the complexity of the surface roughness as depicted in Figure 3. The final dimensions of the model are considered to be 31.1 mm × 11.28 mm × 15 mm. The size of the FE model was determined based on the length of the measured profile, thickness (15 mm), and the width of the studied sample, in such a way to be large enough as a representative model for the real engineering situation.

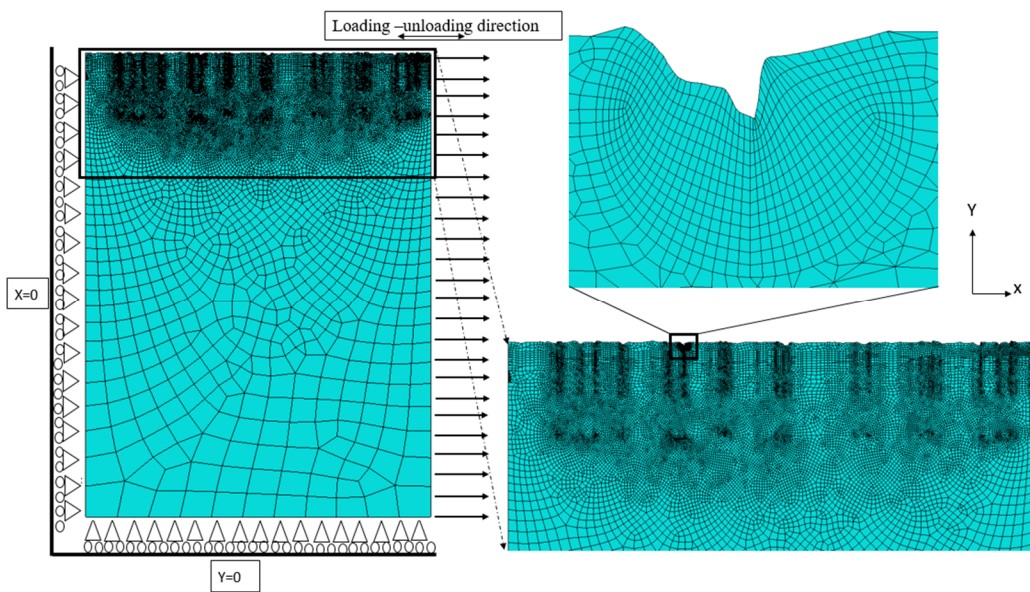

**Figure 3.** Two-dimensional real surface topography model and local mesh [27].

Since the surface roughness consists of different micro-notches with sharp tips in some areas, a free mesh cannot provide accurate results. It is important to use the well-defined mesh especially close to the notch tip. Thus, the mesh element is refined in the micro-defect root to be as small as possible while still being valid for continuum mechanics. The minimum element size is defined as being three times the average grain size of the material, which is equal to roughly 10 μm for the studied steel. With this approach, the material model is valid for a group of grains instead of for an individual grain.

The residual stress effect is considered by proposing layer-wise global modeling with a constant temperature in each layer following the measured residual distribution obtained in the experiments. In Figure 4a, the measurement-based residual stress distribution is shown as a solid smooth line, while the stepwise continuous lines demonstrate the discontinuous residual stress distribution employed in the layer-wise FE model. Figure 4b shows the applicability of the proposed modeling approach in defining layers with constant temperatures using the estimated through-thickness residual stress distribution.

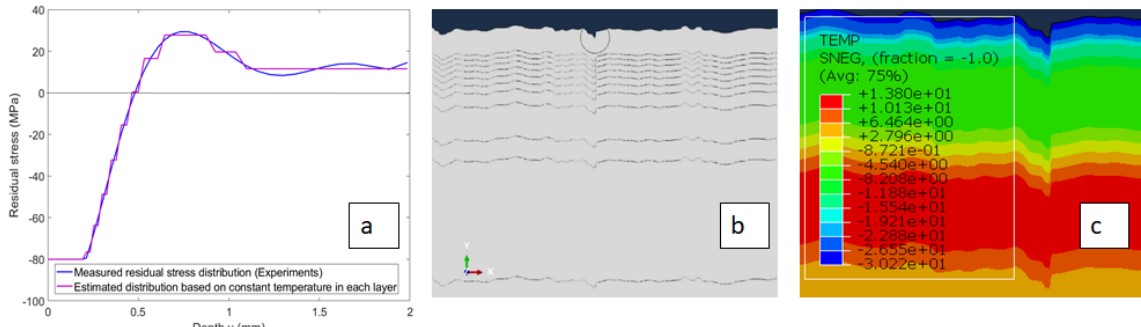

**Figure 4.** (**a**) Continuously approximated residual stress distribution (solid line) and discontinuously approximated residual stress distribution with constant values in each layer (stepwise continuous line) [27], (**b**) layer-wise global modeling, (**c**) the applied temperature field using the proposed approach.

In this method, several layers are defined in the FE model, and the residual stress distributions are presented in the FE analysis as temperature fields [27]. The temperature fields are determined using field distribution, field magnitude, and the thermal expansion coefficient. A typical value of $\alpha = 1.2 \times 10^{-5}$ (1/°C) was used in this study for the thermal

expansion coefficient for steel. The temperature field, applied using the proposed global layer-wise approach, is depicted in Figure 4c for the measured residual stress distribution. In this approach, the temperature field is applied discontinuously, with a constant temperature in each layer following the initial measured distribution. The temperature field is defined as a pre-step, and is applied to the initial undeformed mesh.

The loading condition for the FE simulation is considered to be a single compressive peak load, as shown in Figure 5. The specimen is subjected to compressive loading in order to relax or redistribute the residual stress. The compressive peak load is around the yield stress ($-1.1S_y$), similar to a situation in which a ship is launched or finds itself in severe weather conditions. Then, one tensile load cycle is considered for this material, representing the typical fatigue loading in service.

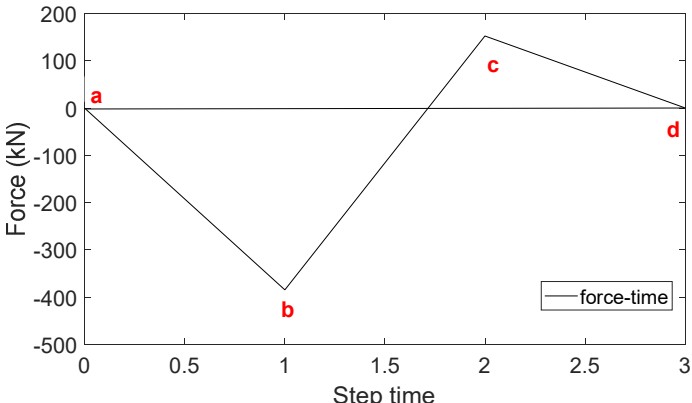

**Figure 5.** Loading condition in one cycle [27].

## 3. Results and Discussion

A sensitivity analysis was carried out for surface roughness, residual stress, and peak load effects. Firstly, the residual stress effect on the damage initiation, evolution, and crack formation was investigated. Moreover, the peak load effect on residual stress distribution and relaxation behavior was studied. Secondly, the influences of peak load and the factors affecting fracture locus were examined.

### 3.1. Influence of Residual Stress on Micro-Crack Formation

Figure 6 shows the damage initiation, evolution, and crack formation based on element deletion for the most critical micro-notch caused by the surface roughness using ductile fracture criteria without the residual stress effect. In Figure 7, the residual stress of $-80$ MPa is also considered in the FEM, using layer-wise global modeling. The result in terms of damage, i.e., formed crack size, is very similar to the analysis cases performed both with and without residual stresses.

After the peak load, the sample is nickel plated, with the nickel layer having a thickness of 20–30 µm, to preserve the surface geometry. Then, sample preparation and polishing are carried out, and the top and the bottom side of the specimen are studied using scanning electron microscopy (SEM). The experimental observations from SEM images after peak load demonstrate the maximum damage size and micro-crack length to be an average of 75 µm; Figure 8a,b. The longest micro-crack on the top surface of the sample is shown with the arrow.

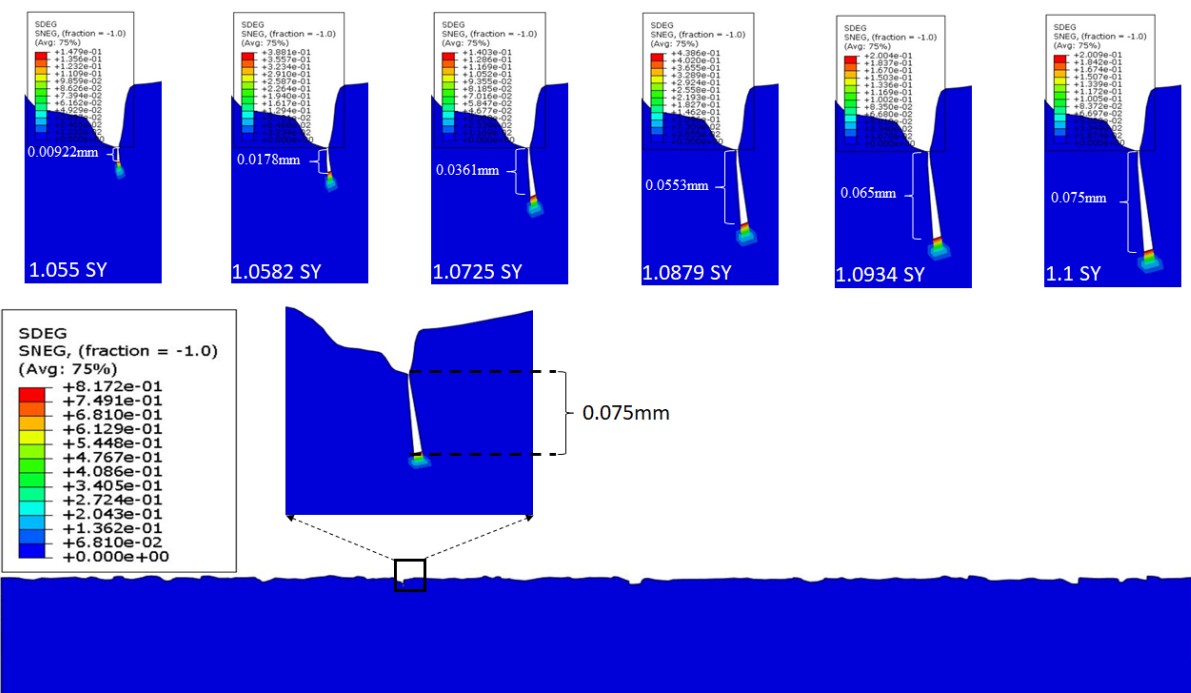

**Figure 6.** Damage initiation, evolution and crack formation for the most critical micro-notch at the surface without residual effect.

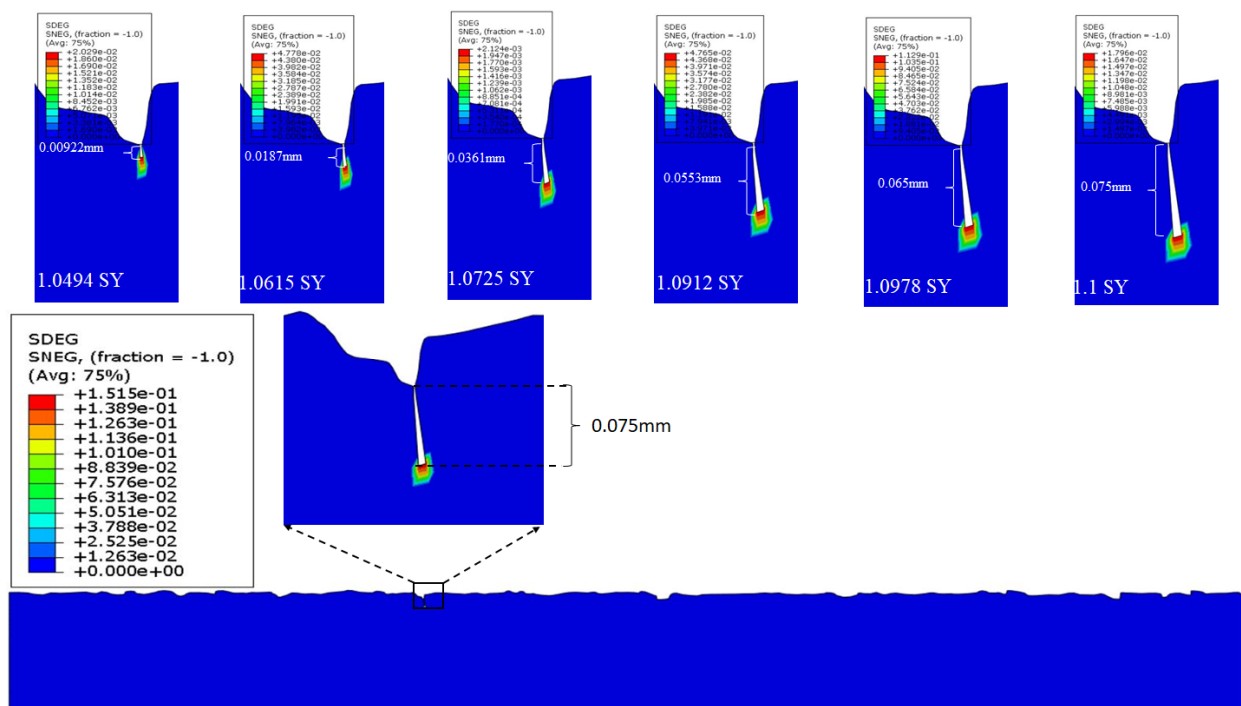

**Figure 7.** Crack formation for the most critical micro-notch at the surface with residual effect, at −80 MPa.

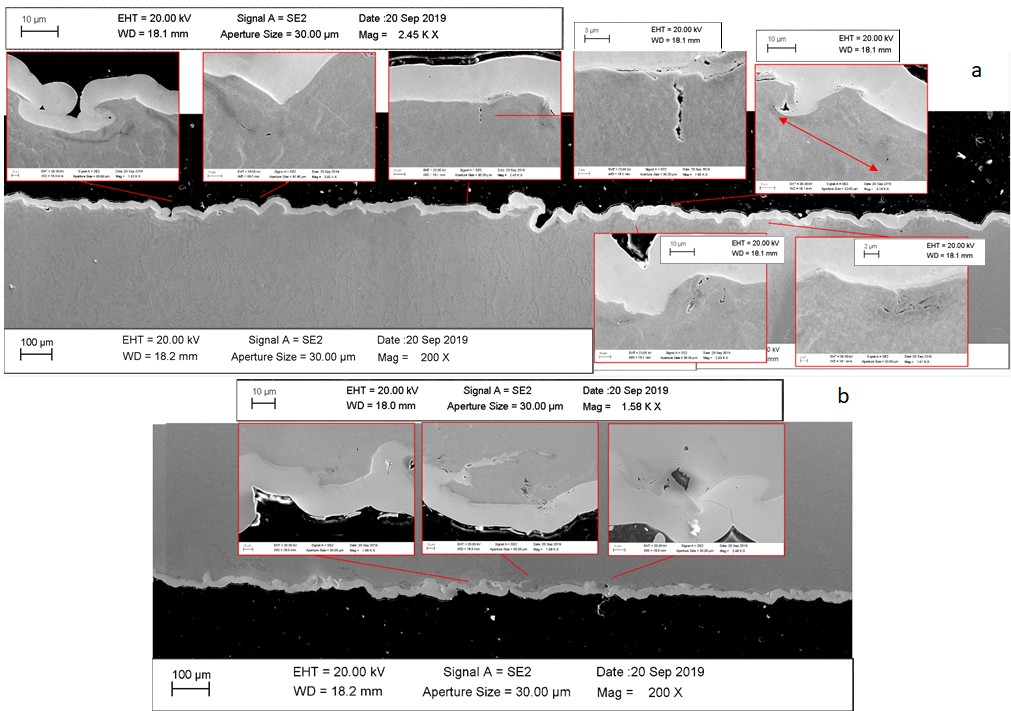

**Figure 8.** The SEM images from (**a**) the top and (**b**) the bottom side of the sample after peak load [27].

The obtained results, as presented in Figures 6 and 7, raise an important question and demonstrate that further understanding and analysis are required in order to determine the effect of residual stress on damage mechanisms. For this purpose, the first element precisely below the critical notch was observed during damage with and without the residual stress effect. Figure 9a shows the stress values versus time step for this element with and without the residual stress effect. The loading time step is considered for the compression peak load from point a to point b, as depicted in Figure 5. In the case in which the residual stress effect is considered, there is a pre-step that we use to define the residual stress. In this figure, the time step from −1 to 0 represents this pre-step. As shown in this figure, the presence of residual stress as defined based on the temperature field in the pre-step causes the element to experience a stress value of −328 MPa at the beginning of the loading step, while in the other case, without the residual stress effect, the stress values in the element start from zero at the beginning of the loading step. It can be seen, at a time step of around 0.3, that the element in both the cases with and without residual stress experiences the same stress values. Figure 9b shows the equivalent plastic strain versus time step for the first element precisely below the critical notch with and without the residual stress effect. If the equivalent plastic strains, which define the failure and removal of the element, are also similar, then we cannot expect differences in the damage initiation and crack formation for these two cases.

This phenomenon might be related to the relaxation of residual stress occurring under this loading condition. The initial residual stress field inherent to or produced during the manufacturing process may change and may not be constant during the operation life of the finished component under residual stress. These residual stresses may be reduced and redistributed, and this diminution is referred to as relaxation. If the sum of the applied and residual stresses locally exceeds the yield stress of a material, residual stress relaxation occurs. Thus, residual stresses do not remain constant, but are relaxed or altered and redistributed during service. Generally, a large relaxation would be expected in the high-stress region, similar to what occurred in this study with an external load value of $-1.1\,S_y$. Figure 10a depicts the stress distribution at different time steps of the loading. It can be seen that the multi-layer residual stress distribution disappears at step $-0.58\,S_y$ of the

loading, or around the middle of the loading step from point a to point b, as depicted in Figure 10b.

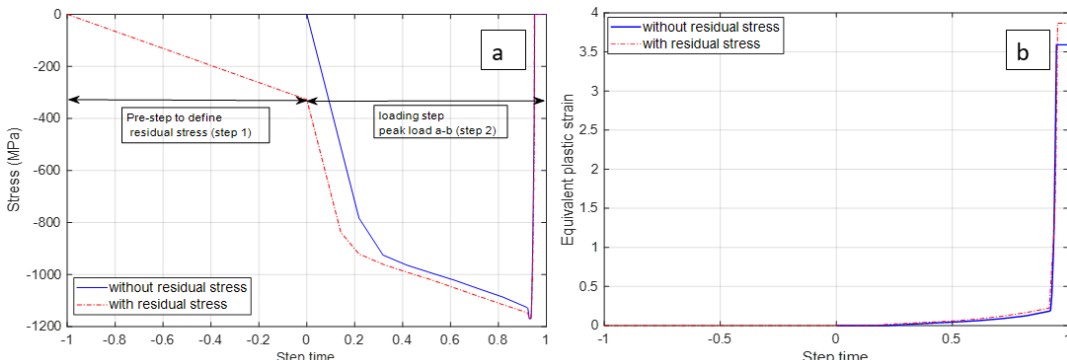

**Figure 9.** (**a**) Stress values versus time step and (**b**) strain versus time step for the first element precisely below the critical notch, without and with a residual stress of −80 MPa.

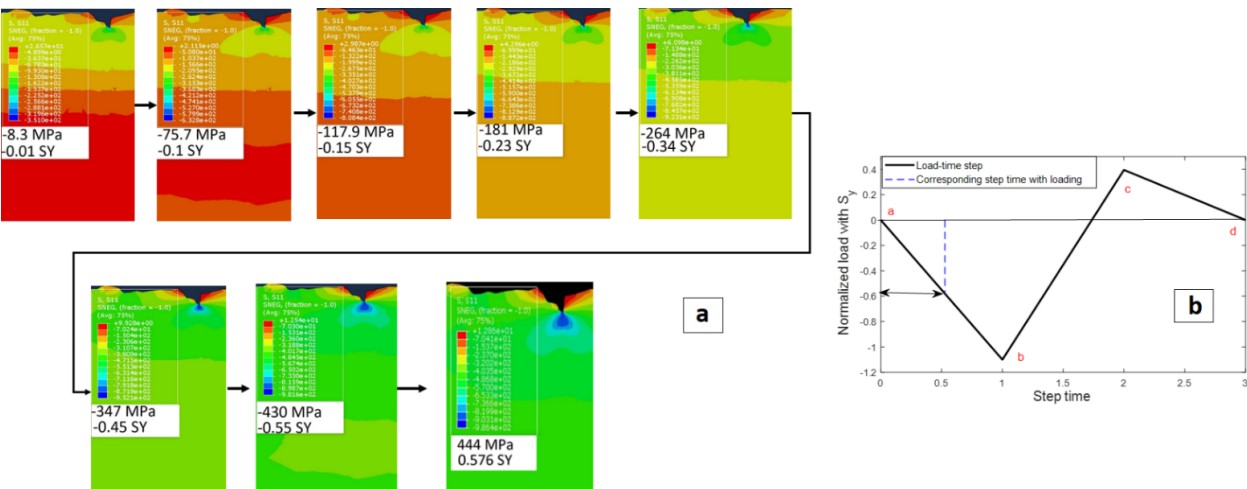

**Figure 10.** (**a**) stress distribution at different time steps of the loading with stress residual effect −80 MPa, (**b**) effective time for residual stress before being relaxed.

For further analysis, the residual stress value is increased from −80 MPa to −320 MPa in order to more precisely investigate the residual stress effect on the damage process. With the increased value of residual stress, it can be seen that there are some differences between the cases with and without residual stress with respect to the length of the crack, as can be seen from a comparison between Figures 6 and 11; however, the crack size does not significantly increase, i.e., the increase is from 75 μm to 84.7 μm. For higher residual stress (−320 MPa), the notch stress value already reaches the yield stress of the material during the pre-step, and the local stress value at the beginning of the loading time step is (for the first element precisely below the critical notch) −947 MPa; see Figure 12.

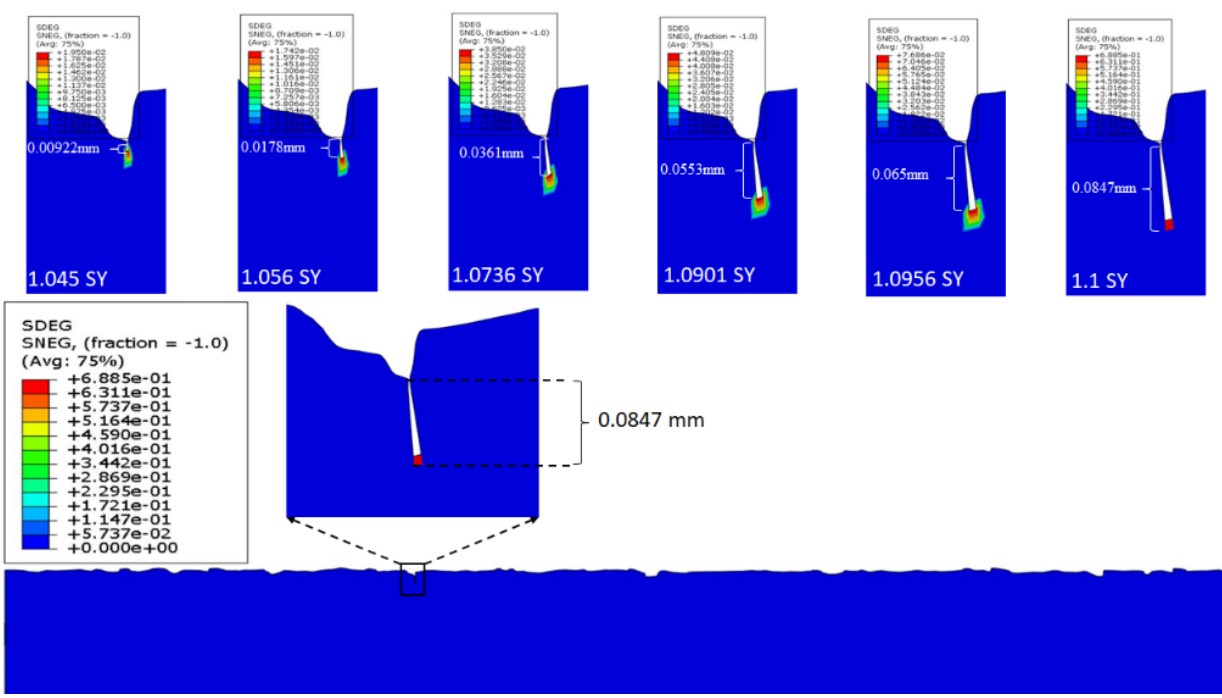

**Figure 11.** Crack formation for the most critical micro-notch at the surface with a residual stress effect of −320 MPa.

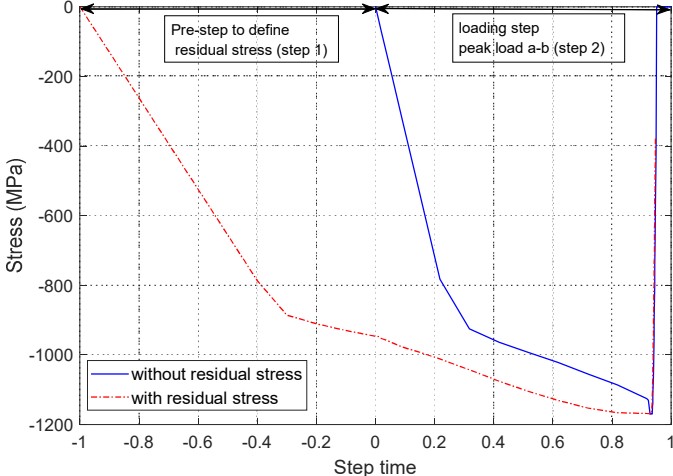

**Figure 12.** Stress values versus time step for the first element precisely below the critical notch without and with residual stress of −320 MPa.

The stress distribution at different time steps of loading for the case of a residual stress of −320 MPa is shown in Figure 13. It can be observed that the global multi-layer residual stress distribution remains effective at step 9 of loading, corresponding to −0.45 $\sigma_y$ of loading; however, it relaxes after some further time steps, corresponding to −1 $\sigma_y$ of loading. It can be concluded that the residual stress relaxation rate, in this case, is lower in comparison with the other case with a lower residual stress value (−80 MPa), while the elements do not undergo plastic deformation in the pre-step with the residual stress effect at the beginning of the loading step. These results also clarify the reason for further crack formation in the case with a higher residual stress value, compared to the case with a lower value.

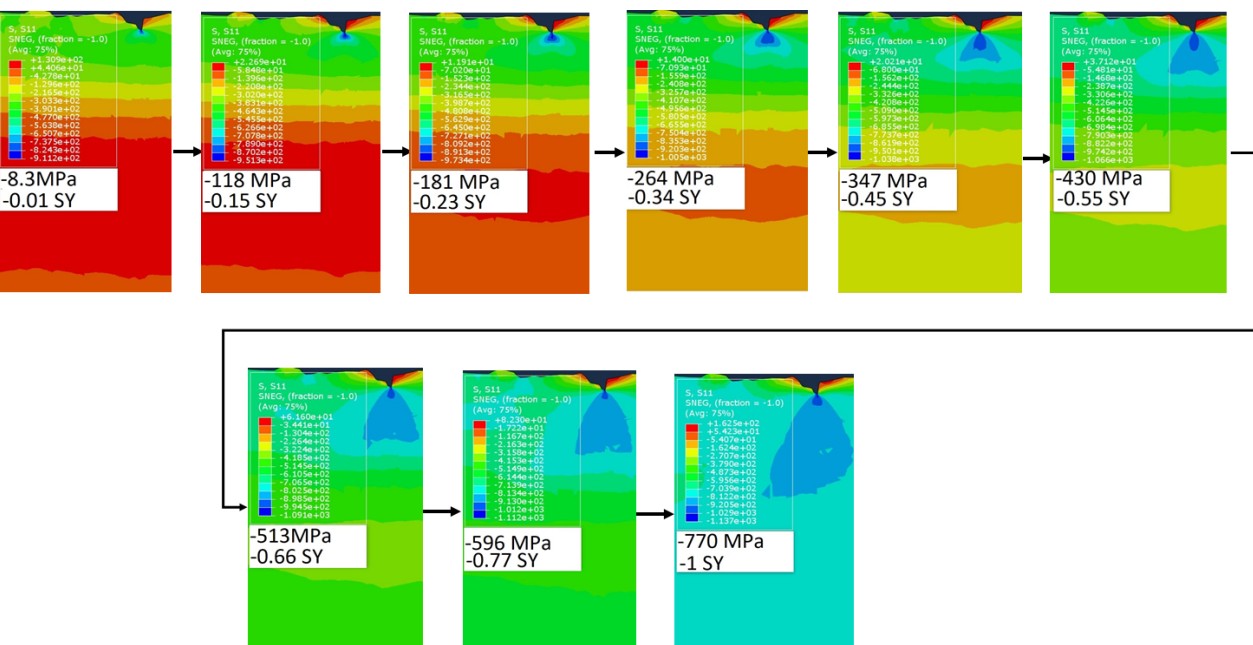

**Figure 13.** Stress distribution at different time steps of the loading with stress residual effect −320 MPa.

Figure 14 shows the crack formation (crack length) as a function of normalized stress value with the nominal yield strength of the material with different residual stress values, i.e., −80 MPa and −320 MPa. The slope of crack length versus load is quite similar for both cases; however, this slope is higher for the residual stress value of −320 MPa at the final step of loading.

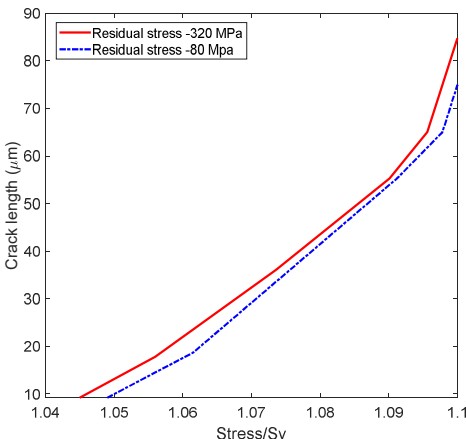

**Figure 14.** Crack length versus normalized stress with the nominal yield strength of the material with different values of residual stress (−80 MPa and −320 MPa).

### 3.2. Influence of Peak Load on Relaxation of Residual Stress

To investigate the influence of peak load values on residual stress relaxation and distribution, further analyses were carried out in cases where there was no overload, while considering the effect of residual stress. To this end, the load history, in this case, was a-c-d, as depicted in Figure 5. Figure 15 shows the stress distribution at different time steps of the loading. It was found that the residual stress remained effective at the end of the loading step; however, it was redistributed during the loading time step. Moreover, the predicted residual stress distribution at the beginning of loading and the stress distribution at the end of the loading step along the path below the critical notch are depicted in Figure 16. The predicted stress exhibits higher values closer to the notch tip

due to the stress concentration factor and the high localized plastic deformation; however, the residual stress remains effective at the end of the loading step without residual stress relaxation. It can be concluded that in the presence of residual stress, the overload value affects the residual stress relaxation rate and distribution.

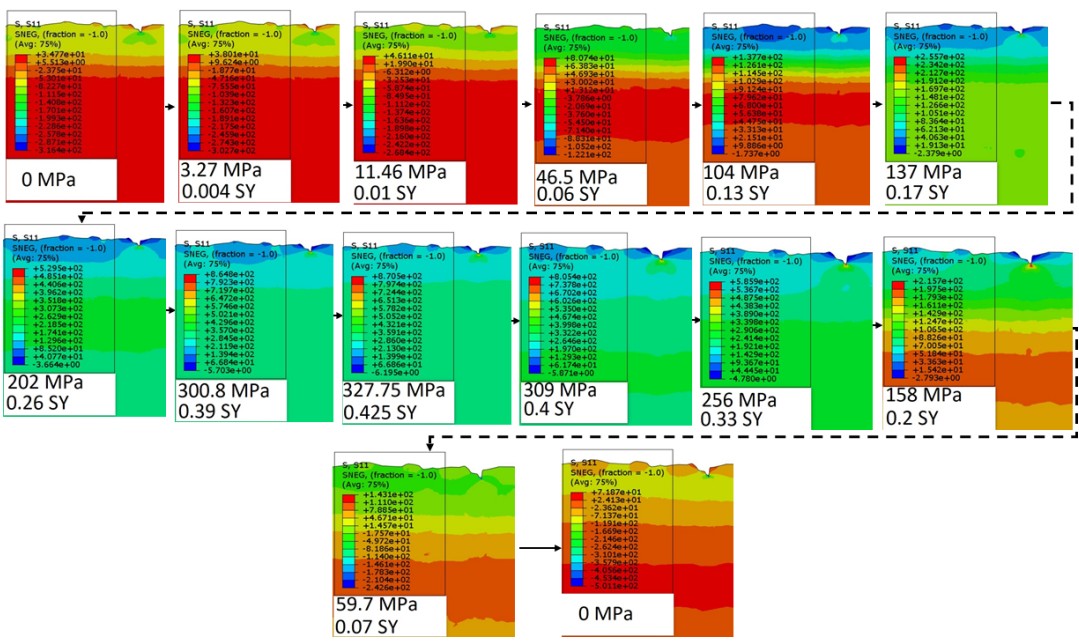

**Figure 15.** The stress distribution at different time steps of the loading without peak load and with the effect of residual stress.

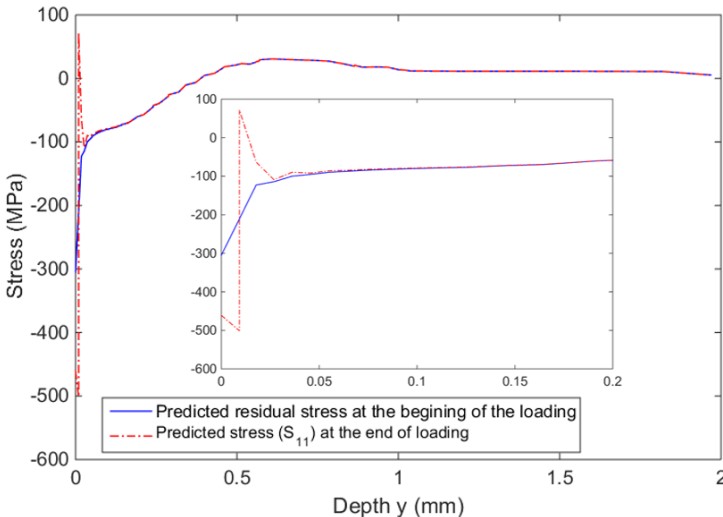

**Figure 16.** The predicted residual stress distribution at the beginning of the loading and the stress distribution at the end of loading without peak load and with the effect of residual.

Further analyses are required to validate these findings; thus, the relaxation of residual stress for different overload values was considered with residual stress values of $-80$ MPa and $-320$ MPa being induced in the material during the manufacturing process. Figure 17a,b shows the stress distribution at different time steps of loading for a peak stress magnitude of $-1.15$ $S_y$ with residual stresses of $-80$ MPa and $-320$ MPa, respectively. For the higher value of residual stress ($-320$ MPa), it can be observed that the residual stress relaxation rate decreased significantly in comparison to the other case with a residual stress effect of $-80$ MPa. Figure 17c shows the duration of the residual stress effect before relaxation at a peak stress magnitude of $-1.15$ $S_y$ with residual stress $-80$ MPa and

−320 MPa. It can be seen for a residual stress of −80 MPa that residual stress relaxation occurs at the time step corresponding to −0.55 $S_y$, while for a residual stress of −320 MPa, the residual stress relaxation occurs later, at the time step corresponding to −0.95 $S_y$.

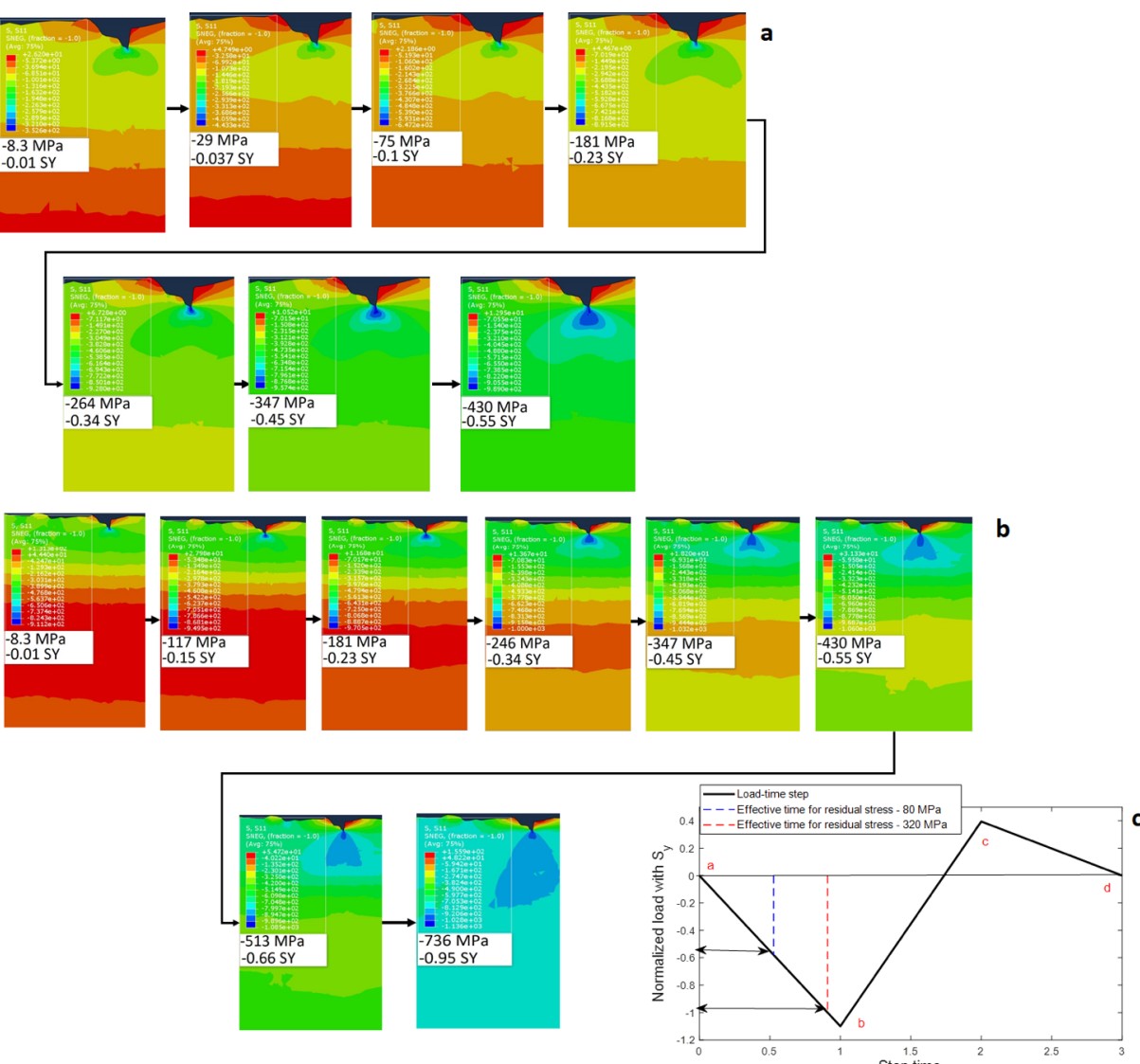

**Figure 17.** The stress distribution at different time steps of loading for an overload magnitude 1.15 $S_y$ before relaxation of the residual stress: (**a**) with residual stress −80 MPa, and (**b**) with residual stress −320 MPa. (**c**) The effective time for different residual stress values before relaxation.

### 3.3. Influence of Fracture Locus and Load Level on Micro Crack Formation

The role of material constants ($C_1, C_2, C_3$) on the ductile fracture criterion has been studied recently [27]. As discussed, $C_2$ is the most significant material constant in the ductile fracture criterion. Parameter $C_2$ determines the equivalent plastic strain to fracture at negative stress triaxiality in the fracture locus, as depicted in Figure 2. In this section, the effect of $C_2$ and overload on crack formation are investigated. The other material constants, $C_1$ and $C_3$, were assumd to be 4 and 1.2, respectively.

Figure 18a,b show the crack size at the most critical notch against different $C_2$ values and their corresponding fracture locus with the overload values −1.1 $S_y$ and −1.15 $S_y$ in cases of two different residual stress values: −80 MPa and −320 MPa, respectively. This figure shows the effect of overload, the $C_2$ parameter (fracture locus), and residual

stress effects on the damage mechanism and crack size, simultaneously. It can be seen that the influence of overload on damage formation and crack size is greater than the residual stress effect, and there are significant differences in the crack size between the overload values $-1.1\ S_y$ and $-1.15 S_y$ at a constant magnitude of residual stress ($-80$ MPa), as shown in Figure 18a. However, there is no significant variance in the crack size between the different magnitudes of residual stress, $-80$ MPa and $-320$ Mpa, with a constant overload value, as depicted in Figure 18b. The relaxation of residual stress is the main reason for this behavior.

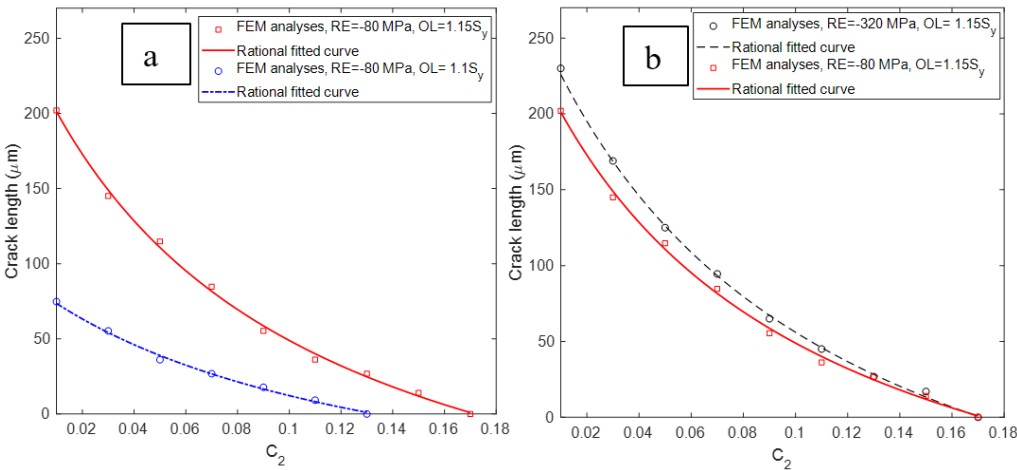

**Figure 18.** Influence of (**a**) overload and (**b**) residual stress on the damage and crack size with the $C_2$ parameter effect.

To further study the effect of the overload, five different magnitudes of peak stress, ranging from $-0.95\ S_y$ to $-1.15\ S_y$, were considered, and the crack sizes for $C_2 = 0.01$ and $C_2 = 0.03$ with a residual stress of $-80$ MPa are presented in Figure 19. It can be seen that there is no damage for overload values lower than $-1\ S_y$, and then, the crack sizes increase exponentially with increasing overload values.

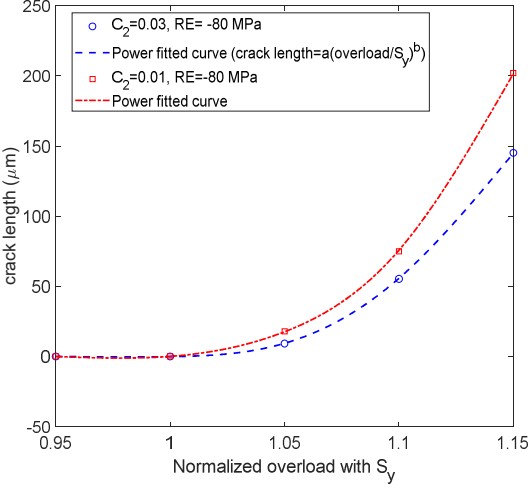

**Figure 19.** Crack length versus normalized overload values (overload/$S_y$) with different values of the $C_2$ parameter: 0.01 and 0.03.

To clarify the effect of surface integrity when the surface is subjected to peak load as a part of loading before other fatigue loading scenarios, the behavior of different elements located close to and far from the crack (see Figure 20a) were studied during the loading time step (see Figure 20b) for three different magnitudes of peak stress, as depicted in Figure 21a–c. It can be seen that the element located close to the crack (element number

2) experienced high stress at time step 3, at the c point of the force-time profile; however, the force magnitude subjected to the element at this stage is low. Thus, this element and the other elements located in the region close to the crack tip are prone to damage even with low magnitudes of loading in other load scenarios after the material has been subjected to peak load. To define the size of the affected region close to the tip of the crack, the stress magnitudes at different locations in the path below the crack tip are depicted in Figure 22a,b for three various overload magnitudes and two different fracture loci, with $C_2 = 0.01$ and $C_2 = 0.03$. This figure shows that the size of the affected region close to the crack tip is more extensive for higher magnitudes of peak stress, especially in cases where the fracture locus has a lower equivalent plastic on at the compression side when $C_2 = 0.01$.

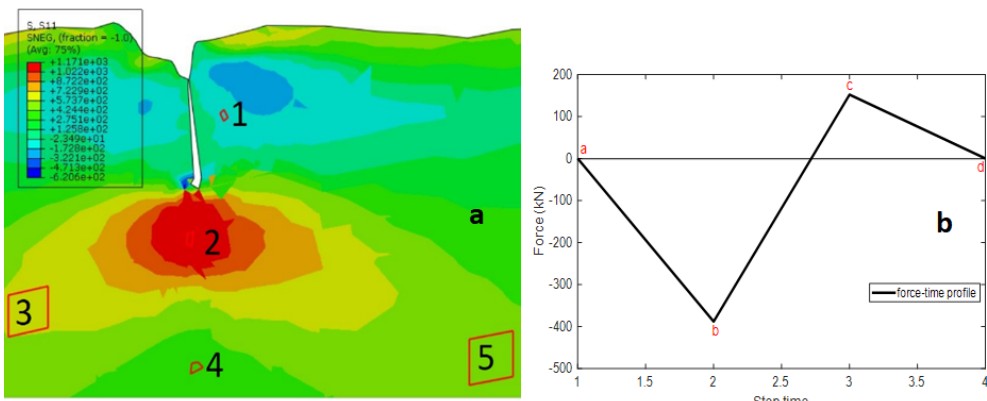

**Figure 20.** (**a**) Locations of five different elements close to and far from the crack, (**b**) loading time profile.

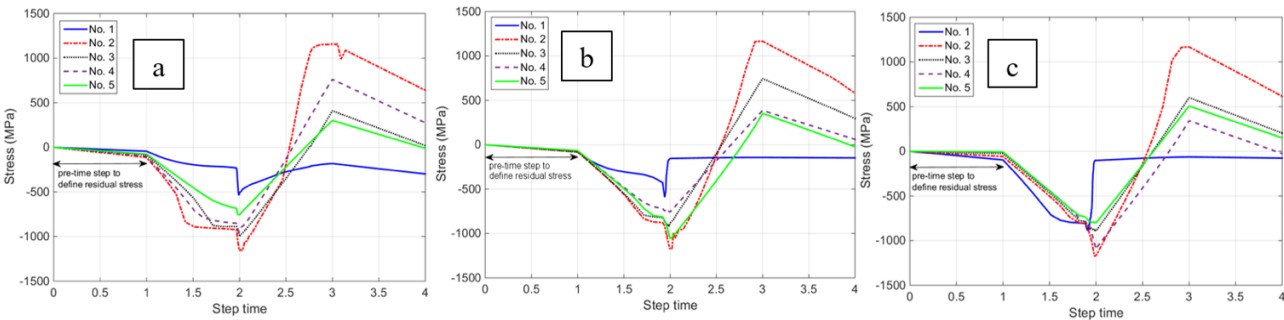

**Figure 21.** Behavior of different elements located close to and far from the crack for three magnitudes of peak stress: (**a**) $-1.05\,s_y$, (**b**) $-1.1\,s_y$, (**c**) $-1.15\,s_y$.

This study additionally establishes new research related to the effect of surface integrity when the peak load is applied to the surface before or during fatigue loading. The existing analytical approach for cut-plate edges, as well as welded joints, uses the original geometry as an initial point for fatigue assessment; meanwhile, the possible influence of peak load on the geometrical parameters, residual stress state, and microcrack formation has been neglected [35–37]. The findings of this study highlight this fact, indicating that considering the original geometry without investigating the possibility of microcrack formation when peak load is a part of the loading scenario can lead to significant inaccuracy in fatigue life prediction. The degree of inaccuracy depends on the peak load magnitude. Since it was found in this study that the crack size increases exponentially with increasing magnitudes of peak load, this neglect could lead to catastrophic problems, especially when engineering components and structures are exposed to high peak stresses and overloads during normal operation and under severe conditions. It should be noted that this study is focused on the numerical investigation of surface integrity, and its influence on microcrack formation of high-strength sandblasted steel under peak load conditions. In future

work, further analysis considering the experimental aspects will be carried out in real engineering situations in order to highlight the importance and effectiveness of the proposed numerical approach.

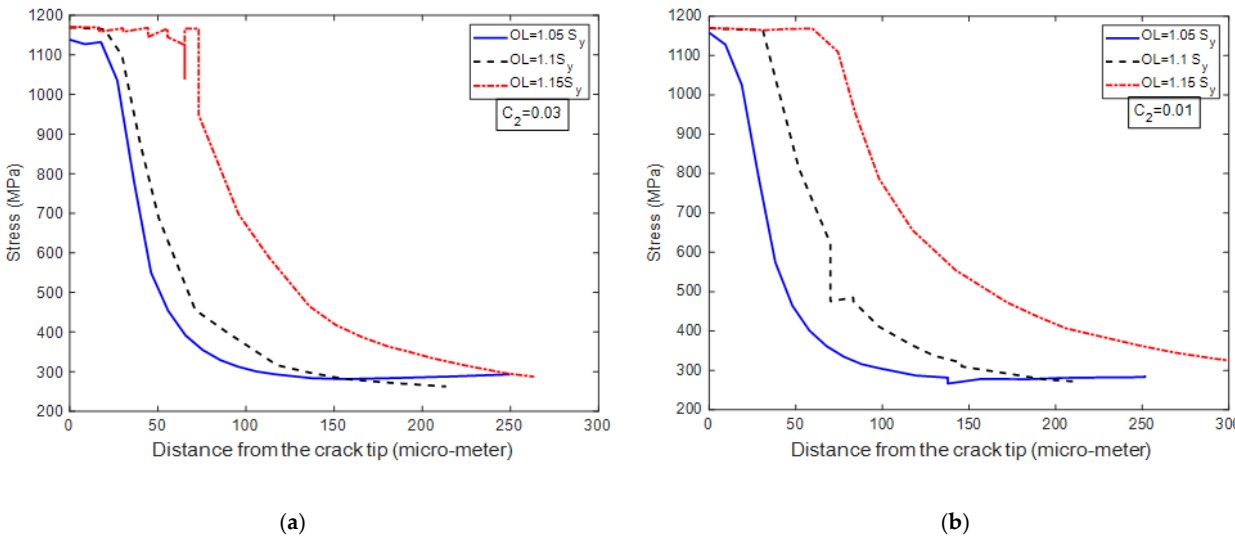

**Figure 22.** Stress distribution along the path below the tip of the crack for (**a**) $C_2 = 0.03$ and (**b**) $C_2 = 0.01$, with a residual stress of $-80$ MPa.

## 4. Conclusions

In this study, a new FEM approach and simulations were employed to characterize the peak load effect on microcrack formation and residual stress state. The numerical simulations simultaneously accounted for the influences of surface roughness and residual stress on the performance of high-strength steel under peak load conditions with ductile fracture criterion. By using this approach, it is possible to monitor and evaluate residual stress relaxation in real time. The main findings of this study can be concluded as follows

- With increasing values of residual stress, the residual stress relaxation rate decreases in comparison to the other studied case with lower magnitudes of residual stress. The residual stress relaxation rate is an affecting parameter with respect to the damage mechanism, and crack size increased with lower residual relaxation rates; however, this effect was not very significant.
- Under peak load conditions, surface roughness has a far more important influence on microcrack formation than residual stresses.
- The influence of compressive overload on damage formation and crack size is greater than the residual stress effect, and there are significant differences in the crack size between various overload values with constant residual stress magnitude. There was no significant variance in crack size between different residual stress magnitudes with a constant overload value. Meanwhile it was found that crack size increases exponentially with increasing magnitudes of peak load in cases with a constant magnitude of residual stress.
- Material areas located in regions close to the crack tip are prone to damage even with low loading magnitudes in other load scenarios after subjecting the material to peak load. The size of the affected region close to the crack tip is more extensive with higher magnitudes of peak stress, especially in cases where the fracture locus has a lower equivalent plastic strain on the compression side.

**Author Contributions:** Conceptualization, J.N.D. and H.R.; methodology, J.N.D.; software, F.S.; validation, J.N.D. and F.S.; Formal analysis, J.N.D.; resources, H.R.; data curation, F.S.; writing—

original draft preparation, J.N.D.; writing—review and editing, J.N.D., H.R., H.H.T.; visualization, H.H.T. All authors have read and agreed to the published version of the manuscript.

**Funding:** This research received no external funding.

**Institutional Review Board Statement:** Not applicable.

**Informed Consent Statement:** Not applicable.

**Data Availability Statement:** Not applicable.

**Acknowledgments:** The present research was supported by project RAMSSES that has received funding under the European Union's Horizon 2020 research and innovation program under the grant agreement No 723246. The information contained herein reflects the views only of the authors, and the European Union cannot be held responsible for any use which may be made of the information contained herein. All financial support is gratefully appreciated.

**Conflicts of Interest:** The authors declare no conflict of interest.

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
