# Peer review of "Influences of Residual Stress, Surface Roughness and Peak-Load on Micro-Cracking: Sensitivity Analysis"

_metals, doi:10.3390/met11020320_

Round 1
Reviewer 1 Report
Language
========
- Minor correction of English should be carried out by the editor,
especially regarding the use of articles and hyphens.
Abstract
========
- The abstract is well written. As it should be, the abstract does not
contains elements of an introduction, but immediately gets to the
point. Also, as it should be, the abstract contains the main
conclusions of the paper.
Introduction
============
- The introduction effectively introduces the topic. Current
challenges are adequately summarized, including references to other
work. Then, the introduction prepares the reader for the
"phenomenological damage model" that this paper uses to improve
current understanding of the influence of compressive peak load on
cracking.
Results and Discussion
======================
- The combination of results and discussion into a single section is
usually not good scientific style.
- The authors should split this section into two separate sections,
one entitled "Results", the other one entitled "Discussion."
- The section "Results" should report the results that other
researchers are supposed to be able to reproduce, objectively and
free of interpretation and opinion. Also, all results should be
reported with corresponding reliability/error limits.
- The section "Discussion" should then show how the authors interpret
these results, how they compare with other results in the
literature, and how they speak to the problem posed in the
Introduction.
- In the current form of the paper, these aspects are not clearly
separated and it is generally difficult for the reader to
discriminate between objective facts and subjective interpretation.
- Splitting this currently combined section into separate Results and
Discussion sections will also reduce each one of those to a more
readable length.
Conclusion
==========
- The conclusion is well-written and actually presents the main
conclusions of the work in a way that exposes broader scientific
impact. The conclusion is not just another summary, as we often get
to see it in less well-written papers.
Equations
=========
- In (2), the "magic" constant numbers, e.g. 3, 1.1015, 1.094 etc
should be replaced by named constants, e.g. k1, k2, k3. Then, after
the equation, the text should be revised to "where ? = 210 MPa, k1 =
3, k2 = 1.1015, k3 = ..."
- Correspondingly in other instances.
Exaggerated Numerical Precision
===============================
- The authors should be aware of the convention that expressions like
"E = 210 MPa" without explicit error limits implicitly mean that the
authors guarantee for the last digit. In other words, "E = 210 MPa"
implicitly means "E = (210.0 ± 0.5) MPa."
- All numbers/quantities in this paper should be checked as to whether
the numerical value properly represents the accuracy to which that
quantity is known (e.g. ±0.5 MPa in the above example). Where ever
this is not the case, proper error/reliability limits should be
indicated.
- Line 256: "stress value around -327 MPa" makes no sense. Why
"around"? "-327 MPa" implicitly means "(-327.0 ± 0.5) MPa." If you
only know this value approximately, e.g. you only know 2 digits,
then you should write "-0.33 GPa." That is the beauty of the SI
system and its prefixes!
SI Units
========
- The paper uses SI units throughout, which is laudable.
Figures
=======
- Please check figures for legibility.
- I cannot read the text at the top of Figure 11.
- Similar in Figure 15, 17, 20, 21.
Other
=====
Line 198
~~~~~~~~
- Unit missing in "α = 1.2×10-5, thermal expansion coefficient for
steel,".
Reviewer 2 Report
- P3, Section 2.1, what is “SFS-EN ISO 4288”? The name of the device should be provided.
- P3, the specific value of the roughness corresponding to Fig.1 should be given.
- P4, the parameters in the constitutive model (Equation 2) should be given clear definitions. And all the simulations in this paper have adopted the constitutive model?
- P4, although the approach to get the material constant C1, C2 and C3 is explained in detail in the previous work, it is necessary to give a general description in this paper. And where is the original data to get these constant?
- P4, the fracture criterion is discussed in detail, but the law of damage evolution is not mentioned.
- P4, why C2 is the most affecting parameter on the micro-crack formation?
- P5, Section 2.3, how to determine the size of FE model as 31.1 mm × 11.28 mm ×15 mm.
- 8, the maximum micro-crack should be marked in this figure.
- The results in Fig.9 and Fig.10 do not seem to correspond to the loading stage(a-b). At the end of the step time, why does the stress increase?
- P9, “It can be seen that the multi-layer residual stress distribution is disappeared at step −0.58 Sy of the loading”, how to get the result that residual stress is disappeared?
- The quality of some figures needs to be improved, such as Fig.7, Fig.10, Fig.13, Fig.17.
- The definitions of some parameters are confusing, such as Sy, σy, and the specific value should be given,
Reviewer 3 Report
Understanding the performance and failure damage mechanism of high strength steels in real engineering applications remain to be a challenge. The influence of surface roughness and residual stress on microcrack formation in high strength steel surface under peak-load condition is looked upon a useful tool to understand the mechanism. Therefore, the current study is on a topic of relevance and general interest to the readers of Metals. Nevertheless, the main concern I have about the paper is with respect to the innovative points. The modelling approach, experimental observation and surface roughness description are very much identical to the work described in cited reference [25]. What is the main strength of the present work submitted to Metals? I believe the authors bear obligation of explaining to the readers.
Small issues:
The authors states: The minimum element size is considered as roughly 10 ??, three times higher than 176 the average grain size of the studied steel. Thus, the material model is valid for a group of grains instead of the individual grain and it follows the principle of continuum mechanics.
What is the principle of defining minimum element size? Is it possible to combine the present approach with CPFE modelling?
What is the unit of the time axis? Is it dimensionless?
The applied temperature field in Fig 4C
How are you supposed to define the failure? micro-crack length to be around 75 μm is still at the stage of stable crack propagation?
Reviewer 4 Report
- It would be good to give the chemical composition of the material tested.
- How does surface roughness affect strength or durability?
- I believe that there are too few references to research supporting the description. Numerical calculations alone do not solve the problem.
- Fig. 8 - invisible description in drawings, e.g. magnification?
- Fig. 19 - the figure shows no overload?
- First sentence in the first conclusion is obvious. I do not agree with the second conclusion, it is just the opposite.
- Author Contributions same XX, YY, ZZ, and where are the authors with these initials?
- Funding, grant number XXX - confidential?
- It would be worthwhile to quote in the introduction also papers of: 1) It would also be worthwhile to quote the following paper: 1) Małecka J., Rozumek D.: Metallographic and mechanical research of the O-Ti2AlNb alloy. Materials Vol. 13, 3006, 2020, 2) Rozumek D., Lewandowski J., Lesiuk G., Correia J., The influence of heat treatment on the behavior of fatigue crack growth in welded joints made of S355 under bending loading. Int. J. of Fatigue 131, 2020.
Round 2
Reviewer 3 Report
The authors have clarified the raised concerns. I would like to recommend its publication on Metals
Reviewer 4 Report
The authors took into account all comments of the Reviewer.